# Association between pre-competition strength and sprint canoe/kayak performance: A mixed-effects analysis of professional Chinese athletes

Zongwei Chen[1,2], Zhaoqian Li[3], Xing Zhang[1], Hongyi Shen[4], Le Yang[4], Birong Lin[4], Xiaoyin Zhang[5], Yangsheng Lin[4]*

1 Department of Physical Education and Sport, Faculty of Sport Sciences, University of Granada, Granada, Spain, 2 School of Physical Education and Sports Science, South China Normal University, Guangzhou, China, 3 School of Physical Education, Shandong University, Jinan, China, 4 Fuzhou Kuiheng Sports Co., Ltd., Fuzhou, China, 5 Beijing Municipal Pollution Source Management Center, Beijing, China

* 553501728@qq.com

## Abstract

This study aimed to examine the associations between pre-competition strength measures and sprint canoe and kayak performance. A post-hoc observational cross-sectional analysis was conducted using official results from the 2023 National Canoe Sprint Autumn Championships (China). Pre-competition strength testing included the one-repetition maximum of the hexagonal deadlift (HDL), bench press (BP), and prone bench pull (PBP), as well as the maximum number of repetitions in the pull-up (PU). Preliminary race results from single-paddler canoe and kayak events were analyzed, including men's 200-m and 1000-m races and women's 200-m and 500-m races. A generalized linear mixed-effects model was used to analyze the data (n = 133; 69 men, 64 women). Fixed effects included relative strength (HDL, BP, PBP, and PU), distance (200 m, 500 m, and 1000 m) and their interactions, sex (male and female), modality (canoe and kayak), and body mass. Athlete identity and heat were treated as random effects. The main findings were: (i) The main model revealed that PBP significantly affected 200 m race time (estimate = −16.7%, $p < 0.001$), but its effect was reduced at 500 and 1000 m (−6% to −1%); (ii) sensitivity analyses indicated that PBP had a greater influence on race time in men (−19.7% to −7.8%) than in women (−10.9% to 2.6%), whereas PU was significant only in women (−0.254%, $p = 0.043$); (iii) PBP had a stronger impact on race time in kayak than in canoe (−25.7% to −8.3% *vs.* −8.6%). Although no strength variables reached statistical significance in canoe ($p = 0.060$–0.239), improvements in PU may still meaningfully contribute to race performance in this modality (95% confidence interval = −0.45% to 0.00935%). Therefore, integrating targeted upper-body resistance training into pre-competition preparation may therefore support improved race outcomes in Chinese professional canoe and kayak athletes.

**Data availability statement:** All relevant data are within the paper and its Supporting Information file.

**Funding:** The author(s) received no specific funding for this work.

**Competing interests:** The authors have declared that no competing interests exist.

## Introduction

Canoe sprint comprises two distinct modalities: canoe and kayak. In a canoe, the paddler competes in a kneeling position using a single-bladed paddle, in contrast to the sitting position and double-bladed paddle used in a kayak. These events are contested over 200, 500, or 1000 meters on flat water, with race durations generally ranging from approximately 35 seconds to 4 minutes [1].

Because high levels of rapid, cyclic, and repetitive force production are required to propel the canoe or kayak, muscular strength is widely regarded as one of the key determinants of competitive performance [2–5]. For example, Kristiansen et al. [5] reported that the bench press (BP) one-repetition maximum (1RM) was a key predictor of 200-m kayaking performance ($R^2 = 0.474$), a finding subsequently validated in a randomized controlled trial in which the training group improved both BP 1RM and 200-m kayaking performance, whereas no improvements were observed in the control group. In addition, Pickett et al. [3] demonstrated that 3RM in the BP, prone bench pull (PBP), and pull-up (PU) were strongly correlated with 200-m kayaking performance ($r = -0.73$ to $-0.80$). Therefore, increasing muscular strength through an organized training program or pre-competition resistance training strategies (*e.g.,* overload and tapering, post-activation performance enhancement, or priming) may contribute to superior competitive outcomes [5–9].

Although most evidence supports a positive relationship between muscular strength and sport performance, some studies have reported contrasting results. For instance, McKean et al. [10] found no significant correlations between 1RM or maximal number of repetitions (MNR) and 500- or 1000-m kayaking performance. It should be noted that previous investigations, regardless of their conclusions, have several limitations. From a statistical perspective, previous studies either failed to account for the nested structure of the data (*e.g.,* repeated observations across distances or modalities) or did not incorporate random effects to control for sources of variability (*e.g.,* individual athlete differences or grouping effects) [4,5,7,11]. From a practical perspective, muscular strength testing and sport performance were sometimes separated by up to two weeks [10], assessed using simulated races or ergometer tests under strictly controlled conditions [4,5], or based on historical official results [7]. Such approaches may introduce temporal mismatches or preclude a direct association between strength measures and sport performance under true competition conditions. In addition, existing studies typically focused on only one race distance, one modality, or a single gender [3–5,7,10]. These issues highlight the need for a comprehensive investigation conducted under official competition conditions to better understand the relationship between pre-competition strength measures and race performance, given that: (i) canoeing and kayaking differ in both biomechanics and energetic demands; (ii) different race distances rely on distinct energy system contributions; and (iii) male and female athletes display well-established differences in muscular characteristics [2].

The primary aim of this study was to examine the relationships between pre-competition strength measures (*i.e.,* hexagonal deadlift [HDL] 1RM, BP 1RM, PBP

1RM, and body-weight PU MNR) and single-paddler sprint canoe and kayak (C1 and K1) race performance in professional male (C1 200 m, C1 1000 m, K1 200 m, and K1 1000 m) and female (C1 200 m, C1 500 m, K1 200 m, and K1 500 m) athletes under official competition conditions [12–14]. We hypothesized that (i) upper-body strength variables, particularly PBP 1RM, are expected to have a stronger influence on kayak performance compared with canoe performance, and (ii) their influence is expected to be greater for shorter distances (200 m) than for longer distances (500 m and 1000 m) [2].

## Materials and methods

### Characteristics of the dataset

The dataset for this study was obtained from the official results booklet of the 2023 National Canoe Sprint Autumn Championship, published by the Chinese Canoe Association (available at http://www.chncanoe.cn/cjfb/2023/1218/591362.html). Use of the dataset was authorized by the Association, and the institutional review board waived ethical approval because only publicly available data were analyzed. All participants were professional canoe or kayak athletes representing various provinces in China and were registered with the Chinese Canoe Association.

The competition took place from December 14–17, 2023. The preliminary race results selected for analysis included the male C1 200 m, C1 1000 m, K1 200 m, and K1 1000 m events, as well as the female C1 200 m, C1 500 m, K1 200 m, and K1 500 m events. On the day before the competition (December 13), the Chinese Canoe Association organized a mandatory strength and conditioning assessment for all athletes. The results of this assessment directly determined athletes' groupings and lane assignments in the official races. Athletes who did not complete all strength tests or who participated in the race competition but did not undergo the strength assessments were excluded from current dataset. The final sample included 69 male athletes (32 C1 and 37 K1) and 64 female athletes (40 C1 and 24 K1) (Table 1).

### Strength testing procedure

Strength testing was conducted from 8:00–17:00 on December 13, 2023. Event officials set up separate testing stations for each strength measure, with each station capable of accommodating multiple athletes simultaneously. Athletes were

**Table 1. Characteristics of body mass, strength measures, and race performance in the athletes.**

| Variable | Canoe | | Kayak | |
|---|---|---|---|---|
| | **Male (n=32)** | **Female (n=40)** | **Male (n=37)** | **Female (n=24)** |
| BM, kg | 78.7±6.7 | 63.4±4.9 | 80.1±6.8 | 64.9±5.0 |
| HDL, kg (Relative to BM) | 149.7±14.1 (1.9±0.1) | 105.6±13.1 (1.7±0.1) | 148.0±15.2 (1.9±0.2) | 104.0±12.4 (1.6±0.1) |
| BP, kg (Relative to BM) | 107.3±17.5 (1.4±0.2) | 66.3±11.4 (1.0±0.2) | 110.8±18.1 (1.4±0.2) | 68.9±13.7 (1.1±0.2) |
| PBP, kg (Relative to BM) | 112.5±15.2 (1.4±0.2) | 77.0±10.9 (1.2±0.1) | 113.2±15.3 (1.4±0.2) | 75.5±11.3 (1.2±0.2) |
| PU, reps | 20.8±7.7 | 16.1±7.5 | 21.2±8.8 | 12.1±7.8 |
| 200 m, s | 47.4±3.2 (30) | 53.5±3.9 (38) | 41.4±2.7 (30) | 47.7±4.2 (16) |
| 500 m, s | / | 143.8±7.3 (29) | / | 127.2±7.0 (21) |
| 1000 m, s | 272.8±14.0 (24) | / | 243.3±10.7 (21) | / |

Values are presented as mean±standard deviation. Values in parentheses for strength measures indicate relative strength, while values in parentheses for race performance indicate the number of athletes in the corresponding modality. Abbreviations: BM, body mass; BP, bench press; HDL, hexagonal deadlift; PU, pull-up; PBP, prone bench pull.

allowed to choose the timing of each test, meaning that the order and intervals between tests were not standardized across athletes. All four exercises (HDL, BP, PBP, and PU) were administered by technical officials in accordance with *the NSCA's Essentials of Strength Training and Conditioning* standards [15]. Specifically, for the 1RM tests of the HDL, BP, and PBP, athletes completed a self-selected warm-up and then performed 5 repetitions at approximately 50% of their self-reported 1RM, 3 repetitions at 70%, and 1 repetition at 90%. The load was subsequently increased in incremental attempts, with a maximum of 5 attempts, to determine the actual 1RM, with 3–5 minutes of rest between attempts. For the PU MNR test, athletes completed a self-selected warm-up and then performed a single set to volitional failure, with the total number of repetitions recorded. Appropriate protective equipment (*e.g.,* weightlifting belts, wrist wraps, and lifting straps) was permitted during testing.

All technical officials (*i.e.,* data recorders and spotters) were affiliated with the Chinese Canoe Association and held a bachelor's or master's degree in sports science. The squat racks, benches, hexagonal barbells, and standard barbells used in testing were from the brand Joinfit (Suzhou, China).

## Race information

The preliminary races were held on December 14 and 16, 2023. Specifically, the 200 m heats were held on December 14 from 14:30–15:15, with a water temperature of 24.4℃, air temperature of 26℃, and a southwest wind of 2.4 m·s$^{-1}$. The 500 m heats took place on December 16 from 08:30–09:05, with a water temperature of 19.9℃, air temperature of 12.1℃, and a north wind of 6 m·s$^{-1}$. The 1000 m heats were conducted on December 16 from 09:20–10:00, with a water temperature of 19.5℃, air temperature of 13.5℃, and a north wind of 4 m·s$^{-1}$.

## Statistical analyses

Statistical analyses were performed using R (version 4.5.2), and the significance level was set at $\alpha = 0.05$. Descriptive data for athletes' strength measures and race performance are presented as mean ± standard deviation. The *glm-mTMB* package was used to fit generalized linear mixed models (GLMMs) with a gamma family and log link to allow assessment of effects and errors in percentage units. Race time was treated as the dependent variable, with athlete ID and heat included as random effects. Fixed effects included HDL 1RM, BP 1RM, PBP 1RM, PU MNR, sex (male *vs.* female), modality (C1 *vs.* K1), distance (200 m *vs.* 500 m *vs.* 1000 m), and body mass. Both absolute strength (kg) and relative strength (relative to BM) were considered, along with interaction terms for strength × sex, strength × modality, and strength × distance. The main model selection was based on the following criteria: (i) lowest Akaike information criterion (AIC) value; (ii) variance inflation factors (VIF) for strength variables < 10; and (iii) satisfactory model diagnostics, which included reasonable residual behavior (assessed via *DHARMa* simulations for uniformity, dispersion, and outliers) and the absence of highly influential cases (verified by a Cook's distance < 0.5) [12,14,16]. For the main model, marginal and conditional $R^2$ values were reported [13], and fixed effects were presented as β coefficients expressed in percentage changes (calculated as % change = [e^β − 1] × 100), along with their standard errors (SEs), 95% confidence intervals (CIs), and *p*-values. To facilitate the practical application of these findings for coaches, the statistical estimates were further converted from proportional changes into absolute performance differences (in seconds). For a specific increment in a strength variable ($\Delta x$), the expected change in race time ($\Delta$time) was calculated using the following equation: $\Delta$time = mean baseline time × (e^{β × Δx} − 1), where the mean baseline time represents the average race duration for each specific sub-group (*i.e.,* by sex, modality, and distance) within the current dataset. Sensitivity analyses were conducted by running the models separately for each sex and for each modality to examine the stability of the predictors. The *ggplot2* package was used to visualize the marginal effects of strength measures on race time and the scatterplots of the observed data.

## Results

Among the eight candidate GLMMs (AIC range: 1162.31–1201.04), the model incorporating absolute strength × distance yielded the lowest AIC but was excluded due to severe multicollinearity (VIF > 10). Consequently, the relative strength × distance model was selected as the main model. It successfully resolved collinearity issues (VIF = 2.14–5.35) while providing the lowest valid AIC (1164.85). Model diagnostics confirmed the robustness of this final specification. Simulated residuals showed no significant deviations from uniformity ($p = 0.323$), no overdispersion ($p = 0.808$), and no significant outliers ($p = 0.342$) (Fig 1). Case-level diagnostics confirmed the absence of influential observations (maximum Cook's distance = 0.075) (Fig 2).

Random effects variances were 1.931e-03 for athlete ID and 1.836e-05 for heat. The model explained 99.5% of the variance through fixed effects (marginal R²) and 99.9% when including random effects (conditional R²). PBP exerted a significant effect on race time at 200 m (% change = −16.7%, $p < 0.001$). However, this effect was markedly attenuated at longer distances, as reflected by the significant PBP × distance interaction at 500 m (15.7%, $p = 0.012$) and 1000 m (10.7%, $p = 0.021$). Effects of strength measures on race time (per 1-unit increase in the measure) were as follows: (i) 200 m: HDL −2.6%, BP −5.86%, PBP −16.7%, and PU −0.0951%; (ii) 500 m: HDL −1.32%, BP −8.85%, PBP −1%, and PU −0.08858%; (iii) 1000 m: HDL −3.12%, BP 6.74%, PBP −6%, and PU −0.2021% (Table 2). Except for BP at 500 m and HDL at 1000 m, all strength variables exerted larger effects on race time at the shorter distance (200 m) than at the longer distances (500–1000 m). The marginal effects and scatterplots illustrating the influence of different strength variables on race time are presented in Fig 2. To facilitate practical application for coaches, the estimated proportional changes in race time were converted into absolute performance differences Fig 3 (in seconds), as presented in Table 3.

Sensitivity analyses were conducted by sex and by modality. In male athletes, PBP significantly influenced race time at 200 m (−19.7%, $p < 0.001$), with a significant PBP × 1000 m interaction (11.9%, $p = 0.012$). In female athletes, PU had a significant effect at 200 m (−0.254%, $p = 0.043$). For modality-specific analyses, PBP was the strongest predictor in K1 at

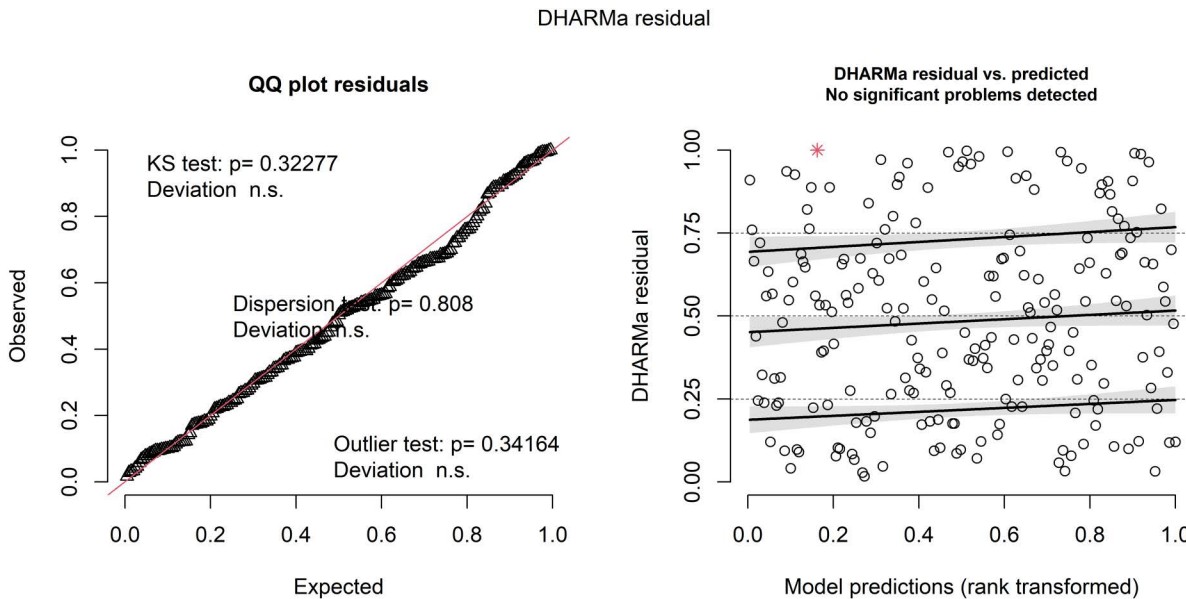

**Fig 1. Residual diagnostics for the main generalized linear mixed-effects model.**

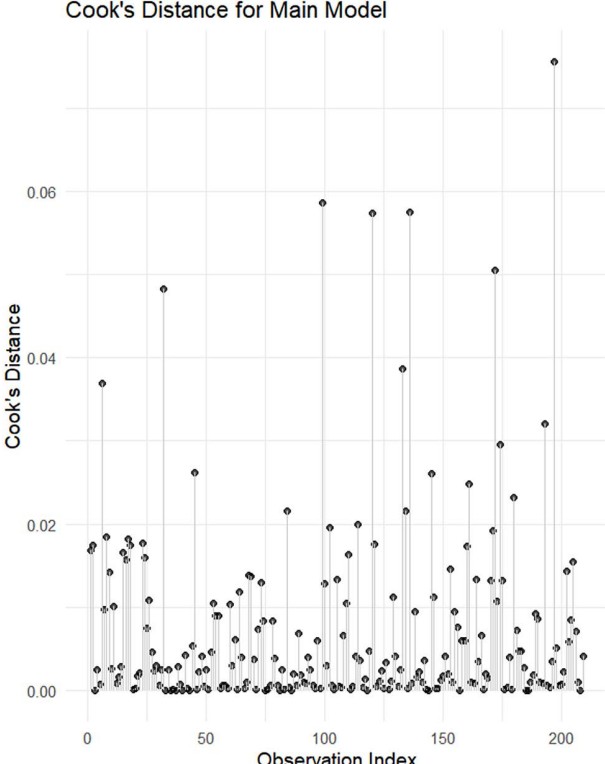

**Fig 2. Cook's distance diagnostics for the main generalized linear mixed-effects model.**

200 m (−25.7%, $p < 0.001$), with a significant PBP × 1000 m interaction (17.4%, $p = 0.017$), whereas no strength variables reached significance in C1 (−8.6% to −0.221%, $p = 0.060$–0.239) (Table 4).

## Discussion

The primary purpose of this study was to examine the relationships between pre-competition strength measures and race performance in professional Chinese canoe and kayak athletes. The main findings of this study were as follows: (i) PBP exerted a significant effect on race time at 200 m; however, this effect was markedly attenuated at longer distances (500–1000 m); (ii) The influence of PBP on race time was greater in men than in women, whereas PU showed a significant effect only in women; (iii) PBP had a stronger impact on race time in K1 compared with C1. Although no strength variables reached statistical significance in C1, improvements in PU may still offer practically meaningful contributions to race performance, given that the upper bound of its 95% CI approaches zero. Therefore, pre-competition strength measures, particularly upper-body strength, may serve as effective predictors of race performance in Chinese professional canoe and kayak athletes.

Previous studies have extensively investigated the determinants of performance in open-water sports such as kayaking, canoeing, rowing, and dragon boat racing. These factors generally encompass anthropometric characteristics, energy metabolism, and physical fitness [2–5,7,10,17]. For example, Ho et al. [17] reported that 200-m dragon boat race performance was significantly correlated with flexed arm girth ($r = 0.57$) and excess postexercise oxygen consumption ($r = 0.57$) measured during the 30 minutes following a 2-minute maximal accumulated oxygen deficit test. In contrast, 500-m

**Table 2. Summary of fixed effects for the generalized linear mixed model (relative strength × distance).**

| Fixed effects | Estimate (% change) | SE | 95% CI | p-value |
|---|---|---|---|---|
| Intercept | 8289 | 671 | (7114, 9656) | **< 0.001*** |
| HDL | −2.60 | 3.11 | (−8.42, 3.58) | 0.402 |
| BP | −5.86 | 3.80 | (−12.90, 1.73) | 0.127 |
| PBP | −16.70 | 4.08 | (−24.20, −8.55) | **< 0.001*** |
| PU | −0.0951 | 0.0765 | (−0.2450, 0.0550) | 0.214 |
| Distance 500 m | 128.0 | 14.9 | (101.0, 158.0) | **< 0.001*** |
| Distance 1000 m | 335.0 | 25.9 | (288.0, 387.0) | **< 0.001*** |
| Sex male | −2.610 | 1.740 | (−5.930, 0.821) | 0.134 |
| Modality K1 | −12.100 | 0.796 | (−13.600, −10.500) | **< 0.001*** |
| BM | −0.1670 | 0.0729 | (−0.3100, −0.0245) | **0.022*** |
| HDL: Distance 500 m | 1.28 | 3.81 | (−5.79, 8.89) | 0.730 |
| HDL: Distance 1000 m | −0.52 | 3.42 | (−6.90, 6.29) | 0.877 |
| BP: Distance 500 m | −2.99 | 5.30 | (−12.60, 7.68) | 0.569 |
| BP: Distance 1000 m | 12.60 | 4.05 | (5.07, 20.70) | **0.001*** |
| PBP: Distance 500 m | 15.70 | 7.49 | (2.26, 30.80) | **0.021*** |
| PBP: Distance 1000 m | 10.70 | 5.02 | (1.53, 20.80) | **0.021*** |
| PU: Distance 500 m | 0.00652 | 0.08370 | (−0.15700, 0.17100) | 0.938 |
| PU: Distance 1000 m | −0.1070 | 0.0864 | (−0.2760, 0.0629) | 0.218 |

* *p* < 0.05. Abbreviations: BM, body mass; BP, bench press; CI, confidence interval; HDL, hexagonal deadlift; K1, single-paddler kayak; PBP, prone bench pull; PU, pull-up; SE, standard error.

performance was significantly correlated with body fat percentage (*r* = −0.55), relaxed and flexed arm girth (*r* = 0.56–0.60), maximal accumulated oxygen deficit (*r* = 0.57), excess postexercise oxygen consumption (*r* = 0.57), and peak power (*r* = 0.57) in elite male Japanese dragon boat athletes. Similarly, López-Plaza et al. [4] found that anthropometric variables such as body mass (*r* = −0.30 to −0.44), body height (*r* = −0.37 to −0.51), sitting height (*r* = −0.40 to −0.64), and maturity status (*r* = −0.55 to −0.73), as well as physical fitness measures such as the overhead medicine ball throw (*r* = −0.44 to −0.62), countermovement jump (*r* = −0.23 to −0.39), and sit-and-reach flexibility (*r* = −0.29 to −0.52), were significantly associated with 200–1000 m canoe and kayak race performance in male Spanish adolescent athletes. Akca et al. [18] further demonstrated that PBP 1RM (*r* = −0.68 to −0.80) was significantly correlated with 200–500 m kayak performance, whereas both the 1-minute PBP and BP repetitions (40% 1RM) (*r* = −0.65 to −0.89) were significantly correlated with 200–1000 m performance in male Turkish national team athletes. Correlations between anthropometric characteristics and race performance appear to be stronger in adolescent athletes than in adults, likely because anthropometric variability decreases with age as athletes at the same competitive level tend to exhibit more homogeneous physiques. For instance, the between-subject coefficient of variation (CV; calculated as standard deviation divided by the mean × 100) for body mass in the present study ranged from 7.7% to 8.5%, whereas in the study by López-Plaza et al. [4], the CV ranged from 12.1% to 17.8%. Considering that metabolic testing is often costly and time-consuming, simple and reliable strength measures may represent practical and effective indicators for large-scale pre-competition assessments [19,20].

Previous studies commonly examining the relationship between strength measures and performance in aquatic sports were conducted under laboratory conditions, where athletes performed strength tests and sport-specific ergometer assessments in a well-rested state [5,9,21]. Some studies have used simulated races under controlled environmental conditions (*e.g.,* water flow, wind speed, and temperature) to obtain performance measures [4,18]. Although such approaches improve the reliability of data collection, they exclude the psychological factors inherent to actual competition, which may

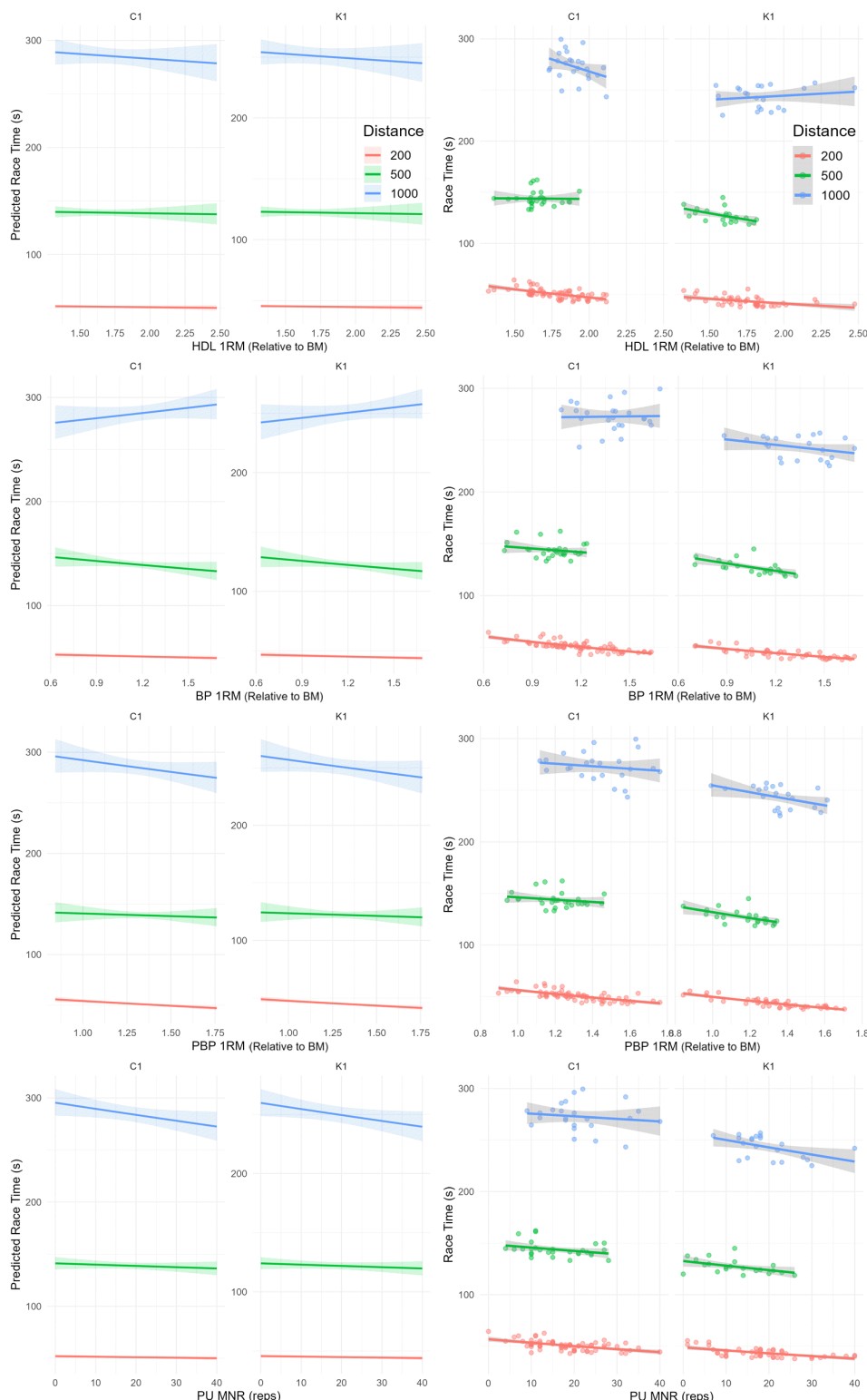

**Fig 3. Marginal effects and fitted scatterplots for hexagonal deadlift (HDL), bench press (BP), prone bench pull (PBP), and pull-up (PU) across modalities (C1 and K1) and distances (200 m, 500 m, 1000 m). Abbreviations: 1RM, one-repetition maximum; BM, body mass; MNR, maximum number of repetitions.**

**Table 3. Estimated changes in race time (seconds) corresponding to a 0.1-unit increase in relative strength (hexagonal deadlift [HDL], bench press [BP], and prone bench pull [PBP]) or a 1-repetition increase in pull-up (PU).**

| Sex | Modality | Distance | Baseline times | HDL | BP | PBP | PU |
|---|---|---|---|---|---|---|---|
| Male | C1 | 200 m | 47.395 | −0.125 | −0.285 | −0.860 | −0.045 |
| | | 1000 m | 272.792 | −0.860 | 1.596 | −2.206 | −0.550 |
| | K1 | 200 m | 41.430 | −0.109 | −0.249 | −0.752 | −0.039 |
| | | 1000 m | 243.284 | −0.767 | 1.423 | −1.967 | −0.490 |
| Female | C1 | 200 m | 53.499 | −0.141 | −0.322 | −0.971 | −0.051 |
| | | 500 m | 143.841 | −0.195 | −1.299 | −0.127 | −0.542 |
| | K1 | 200 m | 47.672 | −0.125 | −0.287 | −0.045 | −0.865 |
| | | 500 m | 127.202 | −0.173 | −1.149 | −0.113 | −0.480 |

Values represent the expected absolute reduction or increase in race time (in seconds), calculated from the optimal Gamma generalized linear mixed model (log-link) incorporating main effects and strength × distance interaction terms. Baseline times represent the mean race duration for each sex-modality-distance subgroup. Negative values denote a reduction in race time (*i.e.,* performance improvement). Abbreviations: C1, single-paddler canoe; K1, single-paddler kayak.

**Table 4. Sensitivity analysis of strength fixed effects on race time.**

| Parameter | Estimate (% change) | SE | 95% CI | *p*-value |
|---|---|---|---|---|
| **Male** | | | | |
| HDL | −1.14 | 4.08 | (−8.67, 7.02) | 0.778 |
| BP | −4.73 | 4.48 | (−12.90, 4.24) | 0.292 |
| PBP | −19.70 | 4.65 | (−28.00, −10.30) | **< 0.001\*** |
| PU | 0.0122 | 0.0949 | (−0.1740, 0.1980) | 0.898 |
| BP: Distance 1000 m | 12.40 | 4.07 | (4.82, 20.50) | **0.001\*** |
| PBP: Distance 1000 m | 11.90 | 5.13 | (2.45, 22.10) | **0.012\*** |
| **Female** | | | | |
| HDL | −2.00 | 5.01 | (−11.10, 8.06) | 0.685 |
| BP | −9.23 | 6.57 | (−20.80, 4.10) | 0.166 |
| PBP | −10.90 | 7.76 | (−24.30, 4.96) | 0.168 |
| PU | −0.25400 | 0.12500 | (−0.49900, −0.00758) | **0.043\*** |
| PBP: Distance 500 m | 13.500 | 7.340 | (0.377, 28.300) | **0.043\*** |
| **C1** | | | | |
| HDL | −6.75 | 4.71 | (−15.30, 2.71) | 0.156 |
| BP | −6.36 | 5.38 | (−16.10, 4.47) | 0.239 |
| PBP | −8.60 | 6.31 | (−19.80, 4.18) | 0.178 |
| PU | −0.22100 | 0.11700 | (−0.45000, 0.00935) | 0.060 |
| BP: Distance 1000 m | 16.40 | 5.63 | (6.12, 27.70) | **0.001\*** |
| **K1** | | | | |
| HDL | 1.25 | 4.11 | (−6.34, 9.45) | 0.755 |
| BP | −5.18 | 5.48 | (−15.10, 5.85) | 0.343 |
| PBP | −25.70 | 5.08 | (−34.80, −15.40) | **< 0.001\*** |
| PU | 0.0572 | 0.1010 | (−0.1400, 0.2550) | 0.571 |
| PBP: Distance 1000 m | 17.40 | 8.15 | (2.96, 34) | **0.017\*** |

Only strength variables and significant interaction terms are presented for the sensitivity analysis models. * *p* < 0.05. Abbreviations: BP, bench press; C1, single-paddler canoe; CI, confidence interval; HDL, hexagonal deadlift; K1, single-paddler kayak; PBP, prone bench pull; PU, pull-up; SE, standard error.

lead to an overestimation of the influence of strength measures on race performance. McKean et al. [10] investigated the association between strength measures and 500- and 1000-m kayak performance under official race conditions at the Australian National Championships. However, several limitations remained: (i) the strength assessments were conducted two weeks before the competition. Given that targeted pre-competition training programs (*e.g.,* overload and tapering) are widely implemented across training squads, athletes' neuromuscular capacities during competition may differ substantially from those assessed two weeks earlier [22,23]; and (ii) the analysis was limited to K1 500- and 1000-m events. Because kayaking involves bilateral movements whereas canoeing emphasizes unilateral paddling, conventional strength measures (*e.g.,* BP, PBP, and PU) may differ in their explanatory power for kayak and canoe performance [24]. Furthermore, the increasingly popular 200-m events were not included [1].

A notable limitation of previous studies examining the effects of strength measures on sport performance is the use of stepwise linear regressions without hierarchical control, which may have inflated observed associations (*e.g.,* Ho et al. [17], López-Plaza et al. [4], and Akca et al. [18]). Another important limitation is that individual- and group-level factors may influence performance outcomes; for example, environmental conditions may vary across heats, and individual differences among athletes may also contribute to variability in race time. These considerations highlight the necessity of using GLMM, which simultaneously accounts for strength measures and random effects associated with athletes and heats. By properly modeling the nested structure of the data, GLMM provides more conservative and reliable estimates of predictor–performance relationships, while preserving the potential to generalize findings to similar athlete populations [12–14]. In the current study, the exceptionally high marginal (99.5%) and conditional (99.9%) $R^2$ inherently result from the model's design: the 'distance' covariate absorbs the massive scale variance across events, the log-link Gamma specification captures proportional scaling, and the random effects account for individual baseline stability.

Consistent with previous findings, upper-body muscular strength appears to be a more important determinant of race performance than lower-body strength, with PBP 1RM emerging as the most influential predictor among the strength measures [3,4,18]. Notably, Akca et al. [18] reported that PBP 1RM significantly correlated with 200-m and 500-m kayaking performance during simulated races, but not 1000-m performance. Our results confirm this distance-dependent attenuation in actual official pre-competition settings, explicitly capturing it through a significant strength × distance interaction. This pattern aligns with the physiological distinction between events: shorter races depend more on rapid force production, whereas longer distances place greater demands on metabolic efficiency [17]. Body mass also emerged as a significant predictor, which is likely related to its contribution to absolute force generation and propulsion in paddle strokes [25]. The effect of PBP differed between modalities, with a larger impact observed in kayak than in canoe. This discrepancy is consistent with the technical demands of the two modalities, as kayakers apply force bilaterally while canoeists generate propulsion unilaterally, reducing the specificity of bilateral strength tests for canoe [9,24]. Strength measures contributed differently between sexes in the present dataset. Male athletes showed a greater dependence on PBP 1RM, whereas race performance in female athletes was more closely associated with PU MNR. Accordingly, resistance-training programs for Chinese professional canoe and kayak athletes may need to be differentiated by sex and modality to maximise performance outcomes [22]. Overall, among Chinese professional athletes, upper-body muscular strength was closely associated with sprint canoe and kayak performance, suggesting it can serve as a valid predictor of race outcomes. Coaches may therefore implement targeted pre-competition resistance training strategies to enhance upper-body muscular strength (*e.g.,* overload and tapering, linear loading, or priming). Moreover, recently developed velocity-based approaches, such as load–velocity and repetition-to-failure–velocity relationships, can be utilized to monitor training load during these exercises, thereby optimizing strength adaptations and improving force production under submaximal loads [26–30].

This study offers several novel contributions. First, strength measures were assessed 1–3 days before official competitions organized by competition authorities, offering a rare opportunity to examine performance predictors under realistic competitive conditions. Second, the sample included high-level professional athletes from multiple provinces, capturing

diverse training backgrounds and demographic characteristics. Third, the use of GLMM allowed simultaneous consideration of individual differences and heat-level variability, yielding more accurate and generalizable estimates than traditional stepwise regression approaches. However, several limitations of the present study should be acknowledged. First, the testing order for each exercise was not standardized. Second, only preliminary race results were analyzed to ensure adequate overall sample size. Third, although the overall sample size (n = 133) is an advantage of this study, the limited sample sizes within specific subgroups (*i.e.,* sex, modality, and distance) may reduce the stability of the corresponding estimates. Fourth, due to data availability constraints, variables such as athletes' age, training experience, and detailed environmental covariates were not included, and thus these findings should be interpreted with caution. Nevertheless, the authors consider this study to provide the closest approximation to real competition conditions for examining the relationship between strength measures and performance outcomes to date. Future studies could implement pre-competition testing under more controlled conditions organized by official competition authorities, and include additional indicators widely considered to influence performance, such as repetitions to a target lifting velocity, repetitions to failure, and load-velocity relationship variables [10,18,31–33]. Such comprehensive investigations could further clarify the relationship between strength measures and race performance, thereby informing the design of training programs for canoe, kayak, and other on-water sports.

## Conclusions

Using GLMM that accounted for both individual athlete differences and heat-level variation, the present study identified muscular strength, particularly upper-body strength, as a meaningful predictor of race performance in professional Chinese sprint canoe and kayak athletes. Overall, athletes with higher PBP 1RM (expressed relative to body mass) performed better in competition, although its influence diminished as race distance increased. When considering sex, PBP 1RM was more closely related to race performance in male athletes, whereas PU MNR was more relevant in female athletes. When considering modality, PBP 1RM contributed more to kayak performance, whereas PU MNR appeared more relevant for canoe. These findings provide initial evidence derived from real competition settings that pre-competition upper-body strength may hold predictive value for sprint canoe/kayak performance. Integrating targeted upper-body resistance training into pre-competition preparation may therefore support improved race outcomes.

## Supporting information

**S1 File.** Dataset.
(XLSX)

**S2 File.** R code.
(R)

## Acknowledgments

The authors gratefully acknowledge the Chinese Canoe Association for providing the dataset and for their valuable support throughout this study.

## Author contributions

**Conceptualization:** Zongwei Chen, Zhaoqian Li.

**Data curation:** Hongyi Shen, Le Yang, Birong Lin.

**Formal analysis:** Zongwei Chen, Zhaoqian Li.

**Investigation:** Hongyi Shen.

**Methodology:** Zongwei Chen, Zhaoqian Li, Xing Zhang, Yangsheng Lin.

**Project administration:** Yangsheng Lin.

**Software:** Zongwei Chen, Xiaoyin Zhang.

**Supervision:** Yangsheng Lin.

**Writing – original draft:** Zongwei Chen, Zhaoqian Li, Xing Zhang, Yangsheng Lin.

**Writing – review & editing:** Zongwei Chen, Zhaoqian Li, Xing Zhang, Hongyi Shen, Le Yang, Birong Lin, Xiaoyin Zhang, Yangsheng Lin.

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
