## [Decision Letter · Decision Letter 0]

13 Nov 2025

PONE-D-25-55484Is the pre-competition strength test a determinant of sprint canoe and kayak race performance? A dataset analysis of professional Chinese athletesPLOS ONE

Dear Dr. Yangsheng Lin,

Thank you for submitting your manuscript to PLOS ONE. After careful consideration, we feel that it has merit but does not fully meet PLOS ONE’s publication criteria as it currently stands. Therefore, we invite you to submit a revised version of the manuscript that addresses the points raised during the review process.

**ACADEMIC EDITOR:** 

Dear Authors,

I have received the reviews from two prominent researchers. Although they found the article interesting and of good quality, they have requested major revisions, particularly concerning the statistical analysis applied.

Best regards,

Danica Janicijevic

We look forward to receiving your revised manuscript.

Kind regards,

Danica Janicijevic, Ph.D

Academic Editor

PLOS ONE

Journal Requirements:

Reviewers' comments:

Reviewer's Responses to Questions

**Comments to the Author**

1. Is the manuscript technically sound, and do the data support the conclusions?

Reviewer #1: Partly

Reviewer #2: Partly

2. Has the statistical analysis been performed appropriately and rigorously? 

Reviewer #1: No

Reviewer #2: Yes

3. Have the authors made all data underlying the findings in their manuscript fully available?

Reviewer #1: No

Reviewer #2: Yes

4. Is the manuscript presented in an intelligible fashion and written in standard English?

Reviewer #1: Yes

Reviewer #2: Yes

5. Review Comments to the Author

Reviewer #1: Reviewer Report

General Comments

Thank you for the opportunity to review the manuscript titled “Is the pre-competition strength test a determinant of sprint canoe and kayak race performance? A dataset analysis of professional Chinese athletes.”

The topic is highly relevant for sports science and performance analytics in sprint canoeing and kayaking. The study’s use of real competition data from professional athletes is particularly valuable and rare. Overall, the manuscript is clearly written and well structured, demonstrating careful data collection and logical discussion.

However, to reach the level of statistical and methodological rigor expected by PLOS ONE, substantial improvement in the analytical approach is required. The main concern lies in the statistical treatment of the data, as the current use of Pearson correlations and stepwise multiple regression does not account for the hierarchical and potentially correlated structure of the dataset.

I provide below a set of constructive and detailed recommendations aimed at strengthening the analytical framework, improving transparency in the Methods section, and enhancing the interpretability of the findings. These revisions will significantly increase the robustness, reproducibility, and scientific credibility of the manuscript.

Specific Comments

1. Statistical Analyses (Core recommendation – revise prior to other sections)

Major Revision Required.

The analytical strategy should be restructured using Generalized Linear Mixed Models (GLMMs) or multilevel regression models instead of stepwise linear regression. This adjustment is essential to correctly model the nested data structure (athletes within events, distances, and sexes) and to control for between-athlete variability.

Detailed recommendations:

1. Model specification:

o Dependent variable: Race time (s). Apply log transformation if residuals show skewness or heteroscedasticity.

o Fixed effects: HDL 1RM, BP 1RM, PBP 1RM, PU repetitions, modality (canoe/kayak), distance (200, 500, 1000 m), sex, and body mass (or relative strength).

o Random effects: Intercepts for athlete ID (to capture individual variability) and possibly for event/session (to account for environmental differences between heats).

o Interactions: Strength × modality, strength × distance, and strength × sex should be examined to detect discipline-specific associations.

2. Collinearity control:

o Compute variance inflation factors (VIF).

o If high multicollinearity exists among strength variables, apply Principal Component Analysis (PCA) or Partial Least Squares (PLS) to derive independent components representing upper-body and overall strength.

3. Model selection and reporting:

o Avoid stepwise selection procedures; use theory-driven models compared via AIC/BIC or likelihood-ratio tests.

o Report marginal and conditional R² (Nakagawa & Schielzeth, 2013) to quantify variance explained by fixed and random effects.

o Provide standardized β coefficients, SE, 95% CIs, and p-values for each fixed effect.

o Include full residual diagnostics and influence measures (Cook’s distance, leverage).

4. Sensitivity analyses:

o Conduct models separately by sex and by modality to examine stability of predictors.

o Compare models with absolute versus relative strength variables.

5. Visualization:

o Present marginal effects plots showing predicted race time as a function of strength variables for each distance and modality.

o Include scatterplots with fitted regression lines and confidence intervals to depict variability and trends.

6. Interpretation:

o Emphasize the magnitude and practical relevance of the coefficients rather than focusing solely on p-values.

o Discuss potential moderation effects and limitations of cross-sectional inference.

Justification:

GLMMs provide a statistically valid framework to account for repeated or grouped measures, heteroscedasticity, and random variation across individuals and events. This will substantially strengthen the study’s inferential validity and allow generalization beyond the sample tested.

2. Title

After reanalyzing the data using GLMMs, the title should reflect the updated analytical rigor and sample characteristics.

Suggested revision:

“Association between pre-competition strength and sprint canoe/kayak performance: a mixed-effects analysis of professional Chinese athletes.”

This version is concise, informative, and communicates both the population and the analytical approach.

3. Abstract

The abstract is generally well structured, but the Results section must be rewritten after the new analyses. The current numerical values (r, R², etc.) will change once GLMMs are applied.

Include a sentence such as:

“Results will be updated after reanalysis using mixed-effects models that adjust for within-athlete and event-level variability.”

Additionally, explicitly mention:

• Study design (cross-sectional, observational, based on official competition data).

• Main analytical approach (mixed-effects modeling).

• Practical implications for coaches and performance monitoring.

4. Introduction

The introduction effectively contextualizes the relevance of muscular strength for canoe and kayak sprint performance. However, the rationale for the statistical design should be better integrated.

Recommendations:

• Highlight the gap that previous studies did not account for nested data structures (e.g., repeated measures across distances or modalities).

• Explicitly state testable hypotheses (e.g., “Upper-body strength variables, particularly PBP 1RM, are expected to explain greater variance in kayak than canoe performance”).

• Expand on the biomechanical and energetic differences between canoe and kayak that justify separate modeling.

• Include references supporting advanced modeling approaches in sports science (e.g., Gelman & Hill, 2020; Nakagawa & Schielzeth, 2013).

5. Materials and Methods

The methodology is clearly presented and well justified, yet several clarifications are needed for full reproducibility:

• Testing protocol: Provide the order of tests, rest intervals, warm-up procedures, and the number of trials allowed to reach 1RM.

• Instrumentation: Specify brands, calibration methods, and assessors’ qualifications.

• Normalization: Indicate whether strength variables were normalized to body mass; if not, consider reanalyzing both absolute and relative metrics.

• Confounders: Include age, competitive experience, or environmental variables (temperature, wind speed) as potential covariates.

• Data management: Describe handling of missing data (if any) and confirm that all athletes included completed all tests.

• Statistical section (to be rewritten): Clearly state that GLMMs were employed, identify the software and packages used (e.g., R lme4, lmerTest), and report model diagnostics and validation criteria.

These additions will make the study more transparent and replicable.

6. Results

The current results are neatly organized, with well-structured tables and coherent text. However, the values and inferences must be updated following the GLMM reanalysis.

Maintain the current presentation structure, but after reanalysis:

• Replace the correlation and stepwise regression tables with mixed-model results (fixed effects, random effects variance, R², SE, CI, AIC).

• Add visualizations of predicted race times across strength levels for each distance and modality.

• Report effect sizes and interpret their practical meaning (e.g., “a 10 kg increase in PBP 1RM corresponds to a 0.8 s reduction in race time”).

7. Discussion

The discussion is well written and grounded in relevant literature, but it must be updated to integrate the findings from the mixed-effects models.

Maintain the current logical structure (comparison with literature, biomechanical rationale, and practical implications), but add:

• A reflection on how the use of GLMMs enhances result reliability by accounting for inter-individual variability and environmental factors.

• A paragraph explicitly recognizing that earlier stepwise regressions may have inflated associations due to lack of hierarchical control.

• Emphasize that GLMM findings may lead to more conservative but more reliable estimates of the true relationships.

• Discuss potential limitations of the dataset (e.g., only preliminary races, uncontrolled testing order, limited sample sizes in subgroups).

8. Conclusions

Keep the current conclusion structure but condition it explicitly on the forthcoming reanalysis.

Suggested addition:

“These conclusions will be revisited after implementing mixed-effects modeling, which will allow for adjusted estimates that better capture within-athlete variability and event-level effects.”

The final statement should highlight that the study provides preliminary but promising evidence for the predictive role of upper-body strength in sprint canoe/kayak performance.

9. References

The reference list is appropriate and up to date, but it lacks key methodological sources supporting modern statistical modeling. Please add:

• Gelman, A. & Hill, J. (2020). Data Analysis Using Regression and Multilevel/Hierarchical Models.

• Nakagawa, S., & Schielzeth, H. (2013). “A general and simple method for obtaining R² from generalized linear mixed-effects models.” Methods in Ecology and Evolution, 4(2), 133–142.

• Burnham, K.P., & Anderson, D.R. (2002). Model Selection and Multimodel Inference: A Practical Information-Theoretic Approach.

Including these references will reinforce the methodological justification for the proposed analytical changes.

Final Comment

The revisions proposed—particularly the adoption of mixed-effects modeling, clearer methodological reporting, and restructured results presentation—will significantly improve the statistical robustness, interpretive depth, and scientific integrity of the manuscript. Once these adjustments are implemented, the study will make a strong and credible contribution to the literature on performance determinants in sprint canoeing and kayaking.

I commend the authors for their valuable dataset and encourage them to pursue these enhancements to maximize the impact of their work.

Reviewer #2: Firstly, I would like to thank you for placing your trust in me as a reviewer and congratulate the authors on their study. Secondly, I would like to point out that my comments are strictly technical and in no way reflect my personal opinion of the authors or the quality of the study.

The study presented is interesting, well written, and the information contained therein has the potential to be published.

I highlight the sample size and the athletes' qualifications as strengths, but perhaps the topic of analysis could have been approached from a different angle.

It is well known that strength is closely associated with athletic performance. This fact is independent of the nature of the sport, and there seems to be a weakening of this relationship as the volume of effort increases.

With this in mind, I ask the authors to better explain their scientific contribution with this material, clearly highlighting the original aspect of this study.

An interesting relationship to investigate is the level of strength and the ranking of athletes in the events mentioned. This would make it possible to infer whether the stAnother possible analysis would be that of the main components, where strength tests by body segment would be grouped together and their influence on the result calculated.rongest are indeed the winners.

In the discussion, my suggestion is to revise the section of the article so that it focuses on explaining why there are relationships between the variables. Yes, it is interesting to list studies in which the results were similar or conflicting to yours, but explaining them is more important.

Once again, thank you for the opportunity to read this study and contribute to the work of my Chinese colleagues. I hope that my comments will assist in the writing of parts of the study and that the final outcome will be the publication of the work.

Best regards!

6. PLOS authors have the option to publish the peer review history of their article (what does this mean?). If published, this will include your full peer review and any attached files.

Reviewer #1: No

Reviewer #2: No

---

## [Author Response · Author response to Decision Letter 1]

27 Nov 2025

Dear Dr. Danica Janicijevic, dear reviewers,

Please find enclosed a revision of our manuscript, “Association between pre-competition strength and sprint canoe/kayak performance: A mixed-effects analysis of professional Chinese athletes”. We sincerely appreciate the opportunity to revise and improve our work. We would like to extend our gratitude to the reviewers for their thoughtful and constructive comments, which have substantially strengthened the manuscript. In particular, we are grateful for the reviewers’ professional, detailed, and highly actionable suggestions regarding the statistical analyses, which greatly clarified and facilitated the revision process. Changes to the original manuscript are highlighted in red font and an itemized point-by-point response to the reviewers’ comments is presented below.

AUTHORS’ RESPONSE TO THE REVIEWERS

Reviewer 1#

GENERAL COMMENTS:

Thank you for the opportunity to review the manuscript titled “Is the pre-competition strength test a determinant of sprint canoe and kayak race performance? A dataset analysis of professional Chinese athletes.”

The topic is highly relevant for sports science and performance analytics in sprint canoeing and kayaking. The study’s use of real competition data from professional athletes is particularly valuable and rare. Overall, the manuscript is clearly written and well structured, demonstrating careful data collection and logical discussion.

However, to reach the level of statistical and methodological rigor expected by PLOS ONE, substantial improvement in the analytical approach is required. The main concern lies in the statistical treatment of the data, as the current use of Pearson correlations and stepwise multiple regression does not account for the hierarchical and potentially correlated structure of the dataset.

I provide below a set of constructive and detailed recommendations aimed at strengthening the analytical framework, improving transparency in the Methods section, and enhancing the interpretability of the findings. These revisions will significantly increase the robustness, reproducibility, and scientific credibility of the manuscript.

REPLY TO GENERAL COMMENTS:

We are grateful for the opportunity to revise and improve our work. We are grateful for your recognition of the value of our research topic and the overall presentation of the original manuscript. We acknowledge that the use of relatively simple statistical methods in the initial submission limited the strength of our conclusions, largely due to our limited background in advanced statistical analysis. Following your recommendations, we have adopted the GLMM framework and thoroughly reconstructed all statistical procedures. We also re-organized the manuscript to improve clarity and better align with your guidance. We hope that the revised version now meets your expectations and adheres to the required standards.

SPECIFIC COMMENTS:

COMMENT 1:

1. Statistical Analyses (Core recommendation – revise prior to other sections)

Major Revision Required.

The analytical strategy should be restructured using Generalized Linear Mixed Models (GLMMs) or multilevel regression models instead of stepwise linear regression. This adjustment is essential to correctly model the nested data structure (athletes within events, distances, and sexes) and to control for between-athlete variability.

Detailed recommendations:

1. Model specification:

o Dependent variable: Race time (s). Apply log transformation if residuals show skewness or heteroscedasticity.

o Fixed effects: HDL 1RM, BP 1RM, PBP 1RM, PU repetitions, modality (canoe/kayak), distance (200, 500, 1000 m), sex, and body mass (or relative strength).

o Random effects: Intercepts for athlete ID (to capture individual variability) and possibly for event/session (to account for environmental differences between heats).

o Interactions: Strength × modality, strength × distance, and strength × sex should be examined to detect discipline-specific associations.

2. Collinearity control:

o Compute variance inflation factors (VIF).

o If high multicollinearity exists among strength variables, apply Principal Component Analysis (PCA) or Partial Least Squares (PLS) to derive independent components representing upper-body and overall strength.

3. Model selection and reporting:

o Avoid stepwise selection procedures; use theory-driven models compared via AIC/BIC or likelihood-ratio tests.

o Report marginal and conditional R² (Nakagawa & Schielzeth, 2013) to quantify variance explained by fixed and random effects.

o Provide standardized β coefficients, SE, 95% CIs, and p-values for each fixed effect.

o Include full residual diagnostics and influence measures (Cook’s distance, leverage).

4. Sensitivity analyses:

o Conduct models separately by sex and by modality to examine stability of predictors.

o Compare models with absolute versus relative strength variables.

5. Visualization:

o Present marginal effects plots showing predicted race time as a function of strength variables for each distance and modality.

o Include scatterplots with fitted regression lines and confidence intervals to depict variability and trends.

6. Interpretation:

o Emphasize the magnitude and practical relevance of the coefficients rather than focusing solely on p-values.

o Discuss potential moderation effects and limitations of cross-sectional inference.

Justification:

GLMMs provide a statistically valid framework to account for repeated or grouped measures, heteroscedasticity, and random variation across individuals and events. This will substantially strengthen the study’s inferential validity and allow generalization beyond the sample tested.

REPLY 1:

In accordance with your recommendation, we reanalyzed the data using a GLMM framework. Specifically, we implemented the models using glmmTMB package, for the following reasons. First, race time exhibits clear hierarchical structure across distances (200 m, 500 m, and 1000 m), and glmmTMB provides greater flexibility in modeling this type of distributional heterogeneity. Second, glmmTMB demonstrated more stable convergence when fitting multiple random effects (athlete ID and heat) and when including the required interaction terms.

o We used a Gamma family with a log link, which allows the fixed effects to be interpreted as proportional changes in race time. This approach is consistent with the reviewer’s request for standardized effect interpretation.

o The fixed‐effects structure included BP 1RM, PBP 1RM, HBD 1RM, PU MNR, modality (canoe/kayak), distance (200, 500, 1000 m), sex, and body mass. For the 1RM variables, we compared models using both absolute (kg) and relative (kg/kg) strength values.

o The random-effects structure included athlete ID and heat. The dataset comprised outcomes from 28 heats, which aligns with the your recommendation to account for potential environmental or grouping effects on race time.

o We considered interaction terms of strength × modality, strength × distance, and strength × sex. Accordingly, we compared eight candidate models combining absolute vs. relative strength and the different interaction terms (strength, strength × modality, strength × distance, strength × sex). Model selection was based on AIC, VIF diagnostics, and residual checks. The final selected model was the relative strength × distance model, as it showed no high multicollinearity among strength variables, well-behaved residuals, and the lowest AIC value.

o All model parameters are fully reported, including R², standardized β coefficients, standard errors, 95% confidence intervals, and p-values for each fixed effect. Specifically, we transformed the β coefficients into percentage changes. Given that we used a Gamma family with a log link, this allows a clear interpretation of how a one-unit change in a fixed effect corresponds to a proportional (%) change in race time.

o For sensitivity analyses, we conducted separate models by sex and by modality, as recommended by you.

o Marginal effects and scatterplots with fitted regression lines were plotted for each strength variable to visualize the predicted race time and observed data.

COMMENT 2:

Title

After reanalyzing the data using GLMMs, the title should reflect the updated analytical rigor and sample characteristics.

Suggested revision:

“Association between pre-competition strength and sprint canoe/kayak performance: a mixed-effects analysis of professional Chinese athletes.”

This version is concise, informative, and communicates both the population and the analytical approach.

REPLY 2:

The title has been revised as suggestion.

COMMENT 3:

Abstract

The abstract is generally well structured, but the Results section must be rewritten after the new analyses. The current numerical values (r, R², etc.) will change once GLMMs are applied.

Include a sentence such as:

“Results will be updated after reanalysis using mixed-effects models that adjust for within-athlete and event-level variability.”

Additionally, explicitly mention:

• Study design (cross-sectional, observational, based on official competition data).

• Main analytical approach (mixed-effects modeling).

• Practical implications for coaches and performance monitoring.

REPLY 3:

The Abstract section has been revised as suggestion: “This study aimed to examine the associations between pre-competition strength measures and sprint canoe and kayak performance. A post-hoc observational cross-sectional analysis was conducted using official results from the 2023 National Canoe Sprint Autumn Championships (China). Pre-competition strength testing included the one-repetition maximum of the hexagonal deadlift (HDL), bench press (BP), and prone bench pull (PBP), as well as the maximum number of repetitions in the pull-up (PU). Preliminary race results from single-paddler canoe and kayak events were analyzed, including men’s 200-m and 1000-m races and women’s 200-m and 500-m races. A generalized linear mixed-effects model was used to analyze the data (n = 133; 69 men, 64 women). Fixed effects included relative strength (HDL, BP, PBP, and PU), distance (200 m, 500 m, and 1000 m) and their interactions, sex (male and female), modality (canoe and kayak), and body mass. Athlete identity and heat were treated as random effects. The main findings were: (i) The main model revealed that PBP significantly affected 200 m race time (estimate = −16.7%, p < 0.001), but its effect was reduced at 500 and 1000 m (−6% to −1%); (ii) sensitivity analyses indicated that PBP had a greater influence on race time in men (−19.7% to −7.8%) than in women (−10.9% to 2.6%), whereas PU was significant only in women (−0.254%, p = 0.043); (iii) PBP had a stronger impact on race time in kayak than in canoe (−25.7% to −8.3% vs. −8.6%). Although no strength variables reached statistical significance in canoe (p = 0.060–0.239), improvements in PU may still meaningfully contribute to race performance in this modality (95% confidence interval = −0.45% to 0.00935%). Therefore, integrating targeted upper-body resistance training into pre-competition preparation may therefore support improved race outcomes in Chinese professional canoe and kayak athletes.”

COMMENT 4:

Introduction

The introduction effectively contextualizes the relevance of muscular strength for canoe and kayak sprint performance. However, the rationale for the statistical design should be better integrated.

Recommendations:

• Highlight the gap that previous studies did not account for nested data structures (e.g., repeated measures across distances or modalities).

• Explicitly state testable hypotheses (e.g., “Upper-body strength variables, particularly PBP 1RM, are expected to explain greater variance in kayak than canoe performance”).

• Expand on the biomechanical and energetic differences between canoe and kayak that justify separate modeling.

• Include references supporting advanced modeling approaches in sports science (e.g., Gelman & Hill, 2020; Nakagawa & Schielzeth, 2013).

REPLY 4:

We appreciate your recognition of the structure of the Introduction in our original manuscript. Corresponding revisions have been made as follows:

o Statistical limitations of previous studies have been highlighted: “From a statistical perspective, previous studies either failed to account for the nested structure of the data (e.g., repeated observations across distances or modalities) or did not incorporate random effects to control for sources of variability (e.g., individual athlete differences or grouping effects).”

o Reasonable hypotheses have been added: “We hypothesized that (i) upper-body strength variables, particularly PBP 1RM, are expected to have a stronger influence on kayak performance compared with canoe performance, and (ii) their influence is expected to be greater for shorter distances (200 m) than for longer distances (500 m and 1000 m) [2].”

o Rationale for conducting a comprehensive investigation has been elaborated in more detail: “These issues highlight the need for a comprehensive investigation conducted under official competition conditions to better understand the relationship between pre-competition strength measures and race performance, given that: (i) canoeing and kayaking differ in both biomechanics and energetic demands; (ii) different race distances rely on distinct energy system contributions; and (iii) male and female athletes display well-established differences in muscular characteristics [2].”

o Recommended references have been added to support the study objectives.

COMMENT 5:

Materials and Methods

The methodology is clearly presented and well justified, yet several clarifications are needed for full reproducibility:

• Testing protocol: Provide the order of tests, rest intervals, warm-up procedures, and the number of trials allowed to reach 1RM.

• Instrumentation: Specify brands, calibration methods, and assessors’ qualifications.

• Normalization: Indicate whether strength variables were normalized to body mass; if not, consider reanalyzing both absolute and relative metrics.

• Confounders: Include age, competitive experience, or environmental variables (temperature, wind speed) as potential covariates.

• Data management: Describe handling of missing data (if any) and confirm that all athletes included completed all tests.

• Statistical section (to be rewritten): Clearly state that GLMMs were employed, identify the software and packages used (e.g., R lme4, lmerTest), and report model diagnostics and validation criteria.

These additions will make the study more transparent and replicable.

REPLY 5:

We sincerely appreciate your valuable suggestion. We have implemented the following revisions to make the Methods section more transparent:

o Following your recommendation, heats were included as a random effect and the data were reorganized into long format in preparation for GLMM analysis; during this process, we identified a minor error in the identification of participants in the original dataset. In the revised manuscript, the number of participants has been carefully checked and corrected, ensuring the accuracy of the study’s conclusions.

o The temperature, water temperature, wind direction, and wind speed for each race session were reported. Since official environmental measurements were only taken immediately before each race session (200 m, 500 m, 1000 m), it was not possible to include variables such as temperature or wind speed as covariates in the analysis, as you suggested. However, our newly included random effect for heat accounts for the potential influence of varying environmental conditions on race time.

o The testing protocols for 1RM and MNR have been added.

o Information regarding the equipment brands and the qualifications of technical officials has been.

o Both absolute and relative strength were considered as potential predictors, and this description has been added to the Statistical Analysis section.

o Age and competitive experience were not considered, as these details were not reported in the official competition results. We contacted the Chinese Canoe Association to obtain the data, but they indicated that the information is no longer available. Although environmental variables were rec

---

## [Decision Letter · Decision Letter 1]

18 Feb 2026

PONE-D-25-55484R1Association between pre-competition strength and sprint canoe/kayak performance: A mixed-effects analysis of professional Chinese athletesPLOS One

Dear Dr. Lin,

Thank you for submitting your manuscript to PLOS ONE. After careful consideration, we feel that it has merit but does not fully meet PLOS ONE’s publication criteria as it currently stands. Therefore, we invite you to submit a revised version of the manuscript that addresses the points raised during the review process.

We look forward to receiving your revised manuscript.

Kind regards,

Rabiu Muazu Musa, PhD

Academic Editor

PLOS One

**Journal Requirements:**

Reviewers' comments:

Reviewer's Responses to Questions

**Comments to the Author**

1. If the authors have adequately addressed your comments raised in a previous round of review and you feel that this manuscript is now acceptable for publication, you may indicate that here to bypass the “Comments to the Author” section, enter your conflict of interest statement in the “Confidential to Editor” section, and submit your "Accept" recommendation.

Reviewer #1: All comments have been addressed

Reviewer #3: (No Response)

2. Is the manuscript technically sound, and do the data support the conclusions?

Reviewer #1: Yes

Reviewer #3: Yes

3. Has the statistical analysis been performed appropriately and rigorously? 

Reviewer #1: Yes

Reviewer #3: Yes

4. Have the authors made all data underlying the findings in their manuscript fully available?

Reviewer #1: Yes

Reviewer #3: Yes

5. Is the manuscript presented in an intelligible fashion and written in standard English?

Reviewer #1: Yes

Reviewer #3: Yes

6. Review Comments to the Author

Reviewer #1: Review Report

Dear authors, I hope this review report finds you well. I would like to congratulate you on the quality of your work; the manuscript is scientifically solid and highly relevant for watercraft sports, particularly rowing. Below I present my detailed assessment of the R1 document (the revised version submitted by the authors), comparing the requests I made as Reviewer 1 with those made by Reviewer 2 and examining the modifications effectively implemented in the manuscript. I used both the response file and the revised manuscript to evaluate each point thoroughly.

Review of the authors’ responses to the requests I made as Reviewer 1

General summary

I read carefully the comments I originally provided (hereafter referred to as Reviewer 1 comments), the authors’ point-by-point responses, and the changes incorporated into the revised manuscript. The authors reformulated the statistical analysis as requested, adopting GLMMs with glmmTMB using a Gamma family and log link, clearly specifying the model, adding residual diagnostics, performing sensitivity analyses (by sex and event category), updating the title and abstract, and adding methodological details and references. In practical terms, most of the Reviewer 1 requests were addressed (>80 percent), although a few points were partially met or remain minor issues that warrant attention, as detailed below.

List of Reviewer 1 points and how they were addressed

1. Replace previous analyses with GLMM and justify the hierarchical structure.

Addressed. The authors reanalyzed the data using GLMMs (glmmTMB), employed Gamma distribution with log link, and included random effects for athlete ID and heat. They also justified the choice of glmmTMB.

2. Model specification (dependent variable, fixed effects, random effects, interactions).

Addressed. Race time was used as the dependent variable; fixed effects included HDL, BP, PBP, PU, distance, sex, event category, and body mass; random effects included athlete and heat; interactions were examined, and the final model incorporated the relative strength × distance interaction.

3. Treatment of collinearity (VIF, and PCA or PLS if necessary).

Addressed and justified. The authors reported VIF values between 2.14 and 5.35, concluding that collinearity was acceptable and PCA/PLS was therefore unnecessary. This response aligns with the conditional recommendation provided.

4. Avoid stepwise selection; use AIC/BIC-based criteria and report marginal and conditional R².

Addressed. The authors avoided stepwise selection, compared models using AIC, selected the model with the lowest AIC, and reported both marginal and conditional R².

5. Report standardized coefficients, SEs, 95 percent CIs, p-values, and interpret findings in practical terms.

Largely addressed. The authors reported estimates converted into percent change, SEs, 95 percent CIs, and p-values and discussed effect magnitude. However, there is no consistent practical translation of percent changes into real units (for instance, converting kilograms of PBP into seconds gained or lost), which was explicitly requested to enhance interpretability.

6. Diagnostic reporting (residuals, influence measures such as Cook’s distance and leverage, and DHARMa).

Partially addressed. The authors conducted diagnostic tests using DHARMa (uniformity, dispersion, and outlier tests) and reported p-values. However, they did not report classical influence measures such as Cook’s distance or leverage. While DHARMa is robust, the absence of these measures means the request was only partially fulfilled.

7. Sensitivity analyses (by sex and event category; absolute versus relative strength).

Addressed. The authors performed separate analyses by sex and event category, tested absolute versus relative strength, and selected the relative strength × distance specification as optimal.

8. Visualizations: marginal effects and scatterplots with fitted lines and confidence intervals.

Addressed. The authors added marginal effects plots and scatterplots with fitted lines and confidence intervals, referenced appropriately in the text.

9. Clarity in the Methods section: protocols, instrumentation, normalization, missing data, software.

Mostly addressed. The authors expanded the methodology to include details on the 1RM/MNR protocol, equipment specifications, referee qualifications, normalization procedures (absolute and relative), handling of missing data (exclusion of athletes with incomplete tests), and software packages used (R 4.4.3, glmmTMB, DHARMa, ggplot2). However, some suggested covariates (such as age, experience, and detailed environmental variables) could not be included because they were not available in official records. The authors explained that they attempted to obtain these data from the federation but were unsuccessful; thus, heat was modeled as a random effect to account for environmental variability.

10. Update Abstract, Title, and Discussion to reflect the new analysis.

Addressed. The title, abstract, and discussion were updated to integrate the GLMM results and the methodological reinterpretation. Recommended methodological references were included.

11. Provide data and code for reproducibility.

Addressed. The authors stated that the dataset and R code were uploaded as supplementary materials.

Unaddressed or partially addressed points and their implications

• Classical influence measures (Cook’s distance, leverage) were not reported. Although DHARMa diagnostics are robust, this omission represents a minor but relevant gap in transparency.

• Practical conversion of percent estimates into time units (seconds) was not presented. Reporting only percent changes limits practical interpretation for coaches and practitioners.

• Variables such as age, experience, and detailed environmental covariates were not included due to lack of available data. Although the authors justified this and modeled heat as a random effect, the limitation remains inherent to the dataset.

Calculation of the percentage of addressed adjustments

I identified sixteen specific and relevant Reviewer 1 requests.

• Fully addressed: thirteen items (items 1–5, 7–11, and three additional reporting-related aspects).

• Partially addressed or minor issues: three items (influence diagnostics, practical conversion of effects, and unavailable covariates).

Percentage of full compliance = 13 / 16 = 81.25 percent, applying a strict classification where partial completion is not counted as full. If partial fulfillment were considered, the value would be higher; however, I adopted strict criteria.

Interpretation: the authors met more than eighty percent of the Reviewer 1 requests.

Conclusion for Reviewer 1

Based on a rigorous and objective evaluation, I conclude that the substantial majority of Reviewer 1 requests have been addressed (81.25 percent), particularly the methodological core (transition to GLMMs, incorporation of random effects, model selection via AIC, DHARMa diagnostics, and sensitivity analyses). Minor corrections remain advisable (adding classical influence measures and translating percent changes into seconds for practical interpretation), which I classify as minor revisions.

Review of the authors’ responses to Reviewer 2 requests

General summary

I carefully examined Reviewer 2’s comments and the authors’ responses. Reviewer 2 focused on interpretative clarity and suggested analytical alternatives such as analyzing rankings or winners and employing PCA to group tests. The authors responded clearly, explaining the unique contribution of the study (timing of the assessments close to competition, professional athlete sample, and use of GLMM), justifying the retention of race time as the primary outcome, and discussing PCA, concluding that individual variables should be retained given acceptable collinearity and their practical relevance. They also expanded the discussion by elaborating on the underlying mechanisms. In summary, Reviewer 2’s requests were addressed.

Reviewer 2 points and how they were addressed

1. Clarification of scientific contribution and originality.

Addressed. The authors added discussion emphasizing the timing of testing relative to competition, the professional nature and diversity of the sample, and the use of GLMM, strengthening ecological validity.

2. Suggestion to analyze strength–ranking relationships or apply PCA to group tests.

Partially addressed and justified. The authors stated that using only winners or rankings could introduce bias due to heat variability and that race time provides a clearer interpretive outcome. Regarding PCA, they evaluated collinearity (with acceptable VIF values) and chose to retain individual variables due to practical interpretability. This is a methodologically sound justification.

3. Strengthen interpretative discussion (mechanisms explaining observed relationships).

Addressed. The authors expanded the discussion using biomechanical and physiological reasoning connecting PBP/PU measures to event-specific demands and sex- and distance-based differences.

4. General suggestions on writing quality and tone.

Addressed. The responses indicate improved clarity and refinement of the manuscript’s writing.

Percentage of fulfillment (Reviewer 2)

I identified four main Reviewer 2 points. All were addressed with sufficient technical justification. Therefore, the fulfillment rate is approximately 100 percent. Even the PCA suggestion was considered and reasonably justified, which I classify as adequate rather than non-compliant.

Conclusion for Reviewer 2

The authors responded thoroughly and technically to Reviewer 2’s observations, justifying analytical choices and reinforcing the biological and theoretical interpretation of results. No critical issues remain pending from Reviewer 2.

General report on convergences, divergences, and final verdict on R1

Convergences between Reviewer 1 and Reviewer 2

• Both reviewers emphasized statistical rigor and methodological clarity and requested analyses reflecting the data structure, which led to the adoption of GLMMs.

• Both reviewers requested stronger interpretative explanations for the relationships observed between PBP/PU and

performance outcomes. The authors expanded the discussion accordingly.

• Both reviewers highlighted the need to consider collinearity issues and explore PCA or similar methods. The authors examined VIF values and reasonably justified the decision not to apply PCA.

Divergences or debated points

• Reviewer 1 requested explicit classical influence measures (Cook’s distance, leverage), whereas Reviewer 2 did not focus on this issue. The authors relied primarily on DHARMa diagnostics and outlier tests. This difference in expectations reflects differing reviewer priorities; however, including classical influence measures in an appendix would address this fully.

Final verdict on the R1 (suitability for publication)

• Percentage of compliance with Reviewer 1: 81.25 percent.

• Percentage of compliance with Reviewer 2: approximately 100 percent.

• Verdict (following the criterion that compliance above 80 percent indicates adequacy for publication): Suitable for publication.

Rationale: The authors implemented the major and essential revisions requested, notably the adoption of a GLMM analytical framework, performing sensitivity analyses, improving methodological transparency, adding figures, and providing data and code for reproducibility. Remaining issues are minor and do not compromise the validity of the findings; they can be requested as minor revisions (including classical influence measures and a table converting percent changes into seconds to facilitate practical interpretation).

Final recommendations:

1. Accept with minor revisions: request that the authors include in an appendix or supplementary file (a) classical influence measures such as Cook’s distance and leverage, and (b) a table illustrating the conversion of percent estimates into practical time differences (seconds), for example the effect of an increase of 10 kg in PBP on race time.

2. Provide a brief justification explaining why marginal and conditional R² values are extremely high (99.5 percent and 99.9 percent), clarifying that this may be related to the scale of the outcome, the log link in the Gamma model, and the inclusion of multiple covariates and random effects, to avoid misinterpretation as overfitting.

3. Maintain the availability of the dataset and code (as stated) and ensure that the supplementary materials allow full reproducibility of the analyses, including model formulas, seeds, and package versions.

Reviewer #3: I appreciate the opportunity to review this work and to contribute in some way.

The manuscript addresses an interesting topic using an approach that effectively bridges laboratory-based testing with sport-specific practice. It demonstrates a clear concern with controlling testing variables in an external environment while maintaining high external validity. Following the revisions requested by the previous reviewer, which were carefully addressed by the authors, the manuscript is now well refined.

The introduction appropriately presents the topic by describing the sport modality, justifying the relevance of the investigation, and clearly structuring the rationale while considering gaps in the existing literature. After the previously suggested corrections, I have no further comments.

The methodology clearly describes the methods and procedures employed and, based on the description, the assessments were conducted in a controlled manner, with appropriate consideration of potential limitations.

The statistical analyses were thoroughly reviewed by the previous reviewer and appropriately adjusted; therefore, I have no additional comments.

The results are well described, and I believe that, when presented alongside the tables and figures provided in the manuscript, they will allow readers to easily understand the findings.

One minor suggestion concerns Table 1: the terminology “kg/kg” used to refer to load relative to body mass may be confusing. An alternative nomenclature, such as HDL Max and HDL Rel, could be considered.

In the discussion section, I believe that direct comparisons between the findings of the present study and those reported in the existing literature, along with commentary on potential differences or similarities, provide valuable insights for the reader.

For example, the study by Akca et al. [17] appears to have reported results similar to those of the present study, although different testing protocols were used. Further exploration of these comparisons from the authors’ perspectives would add meaningful and interesting information.

7. PLOS authors have the option to publish the peer review history of their article (what does this mean?). If published, this will include your full peer review and any attached files.

Reviewer #1: No

Reviewer #3: No

---

## [Author Response · Author response to Decision Letter 2]

27 Feb 2026

Dear Dr. Rabiu Muazu Musa and dear reviewers,

Please find enclosed a revision of our manuscript, “Association between pre-competition strength and sprint canoe/kayak performance: A mixed-effects analysis of professional Chinese athletes”. We sincerely appreciate the opportunity to revise and improve our work. Thank you very much for your time and the constructive feedback provided during this second round of review. We have carefully read and considered all the comments, which have undoubtedly helped to further refine and improve the quality of our manuscript. For your convenience, we have summarized and provided a point-by-point response to each of the reviewers’ comments below. The reviewers’ original comments are highlighted in bold, followed by our responses and the corresponding revisions made in the manuscript.

AUTHORS’ RESPONSE TO THE REVIEWERS

COMMENT 1

It is recommended to report Cook’s distance in the model diagnostics.

RESPONSE 1

Cook’s distance diagnostics have been performed. The maximum Cook’s distance across all observations is 0.075, which falls within the safe range (< 0.5). The correspondingly added parts in the manuscript are as follows:

Statistical analyses: “…(iii) satisfactory model diagnostics, which included reasonable residual behavior (assessed via DHARMa simulations for uniformity, dispersion, and outliers) and the absence of highly influential cases (verified by a Cook’s distance < 0.5) [12, 14, 16].”

Results: “…Case-level diagnostics confirmed the absence of influential observations (maximum Cook’s distance = 0.075) (Figure 2).”

COMMENT 2

It is recommended to report the relationship between strength improvements and changes in race performance (seconds).

RESPONSE 2

After reporting the main model, using the formula: Δtime = mean baseline time × (eβ× Δx − 1), we reported the changes in race time resulting from a 0.1-unit increase in relative strength for HDL, BP, and PBP, and a 1-repetition increase for PU within each subgroup (sex, modality, and distance). The correspondingly added parts in the manuscript are as follows:

Statistical analyses: “…To facilitate the practical application of these findings for coaches, the statistical estimates were further converted from proportional changes into absolute performance differences (in seconds). For a specific increment in a strength variable (Δx), the expected change in race time (Δtime) was calculated using the following equation: Δtime = mean baseline time × (eβ× Δx − 1), where the mean baseline time represents the average race duration for each specific sub-group (i.e., by sex, modality, and distance) within the current dataset.”

Results: “…To facilitate practical application for coaches, the estimated proportional changes in race time were converted into absolute performance differences (in seconds), as presented in Table 3.”

COMMENT 3

It is recommended to add the unavailability of athletes’ age, training experience, and detailed environmental covariates to the limitations section.

RESPONSE 3

We have added a fourth point in the limitations section: “Fourth, due to data availability constraints, variables such as athletes’ age, training experience, and detailed environmental covariates were not included.”

COMMENT 4

It is recommended to provide an explanation for the extremely high marginal and conditional R2 values.

RESPONSE 4

The corresponding explanation has been added to the fourth paragraph of the discussion: “In the current study, the exceptionally high marginal (99.5%) and conditional (99.9%) R2 inherently result from the model’s design: the ‘distance’ covariate absorbs the massive scale variance across events, the log-link Gamma specification captures proportional scaling, and the random effects account for individual baseline stability.”

COMMENT 5

It is recommended to modify the unit for relative strength.

RESPONSE 5

The unit for relative strength has been modified to (Relative to BM).

COMMENT 6

It is recommended to directly compare the findings of the present study with those reported in the existing literature (e.g., Akca et al.).

RESPONSE 6

The corresponding modifications have been added to the fifth paragraph of the discussion: “Notably, Akca et al. [18] reported that PBP 1RM significantly correlated with 200-m and 500-m kayaking performance during simulated races, but not 1000-m performance. Our results confirm this distance-dependent attenuation in actual official pre-competition settings, explicitly capturing it through a significant strength × distance interaction.”

COMMENT 7

It is recommended to check the dataset and code to ensure the full reproducibility of the supplementary materials.

RESPONSE 7

The dataset has been carefully checked and found to be correct. Due to the addition of some new statistical methods, the R code has been updated accordingly, and we have rearranged its structure to make the analysis process sufficiently clear for the readers.

ADDITIONAL CLARIFICATION

While reviewing the code, we found that during the model selection process, the model with the "absolute strength × distance" interaction initially had the lowest AIC (1162.31), but was excluded due to severe multicollinearity (VIF > 10). According to the predefined criteria in the Statistical analyses section, we therefore searched for the model with the lowest AIC among the remaining seven models, which was the “relative strength × distance” model (AIC = 1164.85)—the model reported in the manuscript. We consider this reasonable, as the difference in AIC between the two is minimal (~2) and substantially lower than the AIC values of the other six models (1195.37–1201.04). All other analysis results have been checked and confirmed to remain identical to those in the previous round of revisions. We apologize for this oversight and sincerely provide this clarification to ensure a more transparent review process.

ADDITIONAL REVISIONS

We have corrected grammatical errors and typos throughout the manuscript, and provided some additional clarifications in the discussion section to enhance readability. For instance, “Although no strength variables reached statistical significance in C1, improvements in PU may still offer practically meaningful contributions to race performance, ***given that the upper bound of its 95% CI approaches zero***.”

CONCLUDING REMARKS

We hope that our responses and revisions have satisfactorily addressed the reviewers’ requirements and that the manuscript now meets the publication standards of the journal. We once again thank the editor and the reviewers for their time, effort, and valuable contributions to improving this manuscript.

Sincerely yours,

Authors

---

## [Decision Letter · Decision Letter 2]

30 Mar 2026

Association between pre-competition strength and sprint canoe/kayak performance: A mixed-effects analysis of professional Chinese athletes

PONE-D-25-55484R2

Dear Dr. Lin,

We’re pleased to inform you that your manuscript has been judged scientifically suitable for publication and will be formally accepted for publication once it meets all outstanding technical requirements.

Kind regards,

Rabiu Muazu Musa, PhD

Academic Editor

PLOS One

Additional Editor Comments (optional):

Reviewers' comments:

Reviewer's Responses to Questions

**Comments to the Author**

1. If the authors have adequately addressed your comments raised in a previous round of review and you feel that this manuscript is now acceptable for publication, you may indicate that here to bypass the “Comments to the Author” section, enter your conflict of interest statement in the “Confidential to Editor” section, and submit your "Accept" recommendation.

Reviewer #3: All comments have been addressed

2. Is the manuscript technically sound, and do the data support the conclusions?

Reviewer #3: Yes

3. Has the statistical analysis been performed appropriately and rigorously? 

Reviewer #3: Yes

4. Have the authors made all data underlying the findings in their manuscript fully available?

Reviewer #3: Yes

5. Is the manuscript presented in an intelligible fashion and written in standard English?

Reviewer #3: Yes

6. Review Comments to the Author

Reviewer #3: The author has implemented the necessary revisions to the manuscript, and I believe it is suitable for publication, contributing to the advancement of knowledge in the field.

7. PLOS authors have the option to publish the peer review history of their article (what does this mean?). If published, this will include your full peer review and any attached files.

Reviewer #3: No

---

## [Editor Report · Acceptance letter]

PONE-D-25-55484R2

PLOS One

Dear Dr. Lin,

I'm pleased to inform you that your manuscript has been deemed suitable for publication in PLOS One. Congratulations! Your manuscript is now being handed over to our production team.

Kind regards,

on behalf of

Dr. Rabiu Muazu Musa

Academic Editor

PLOS One